# Recent Tissue Engineering Approaches to Mimicking the Extracellular Matrix Structure for Skin Regeneration

**DOI:** 10.3390/biomimetics8010130

**Published:** 2023-03-22

**Authors:** Rikako Hama, James W. Reinhardt, Anudari Ulziibayar, Tatsuya Watanabe, John Kelly, Toshiharu Shinoka

**Affiliations:** 1Center for Regenerative Medicine, The Abigail Wexner Research Institute, Nationwide Children’s Hospital, 700 Children’s Drive, Columbus, OH 43205, USA; 2Department of Biotechnology and Life Science, Graduate School of Engineering, Tokyo University of Agriculture and Technology, 2-24-16 Naka-Cho, Koganei 184-8588, Japan; 3Department of Cardiothoracic Surgery, The Heart Center, Nationwide Children’s Hospital, 700 Children’s Drive, Columbus, OH 43205, USA; 4Department of Surgery, Cardiovascular Tissue Engineering Program, Ohio State University, Columbus, OH 43210, USA

**Keywords:** skin regeneration, dressing materials, tissue engineering, electrospinning nanofiber, sponge, hydrogel, surface topology, mechanobiology

## Abstract

Inducing tissue regeneration in many skin defects, such as large traumatic wounds, burns, other physicochemical wounds, bedsores, and chronic diabetic ulcers, has become an important clinical issue in recent years. Cultured cell sheets and scaffolds containing growth factors are already in use but have yet to restore normal skin tissue structure and function. Many tissue engineering materials that focus on the regeneration process of living tissues have been developed for the more versatile and rapid initiation of treatment. Since the discovery that cells recognize the chemical–physical properties of their surrounding environment, there has been a great deal of work on mimicking the composition of the extracellular matrix (ECM) and its three-dimensional network structure. Approaches have used ECM constituent proteins as well as morphological processing methods, such as fiber sheets, sponges, and meshes. This review summarizes material design strategies in tissue engineering fields, ranging from the morphology of existing dressings and ECM structures to cellular-level microstructure mimicry, and explores directions for future approaches to precision skin tissue regeneration.

## 1. Introduction

The costs associated with acute and chronic wound care are estimated to range from approximately USD 28 billion up to USD 97 billion as of 2018 [1]. Treatment costs are particularly high for surgical wounds and diabetic foot ulcers. Whereas dressings have been developed to induce acute wound healing, options for chronic wounds are inadequate and skin grafts are still used. Therefore, the financial and caregiver burdens continue to grow as the number of people with diabetes and the elderly population, who are more susceptible to chronic wounds, such as bedsores and diabetic ulcers, continue to increase [2,3]. In addition, restoring the structure and function of the skin has a significant impact on the patient’s quality of life, especially in terms of aesthetics.

Historically, wound dressings (e.g., gauze and bandages) have been used to absorb exudates and act as temporary physical barriers, keeping the wound dry by frequent reapplication [4]. This reapplication process tends to damage the re-epithelialized tissue adhering to the bandage and is a major factor in the chronicity of the inflammatory phase. Therefore, attention has been focused on providing a smooth healing environment and avoiding delays in healing, aiming for a treatment method that takes advantage of the patient’s natural healing ability. In the late 20th century, it became apparent that a moist environment was necessary to keep the wound moist in order to avoid slowing the healing process. Wound dressings are now commonly used to absorb secreted exudate and create a humid environment. This is very important in eliminating desiccation damage to cells and preventing infection from bacteria. Delays in the healing process are mainly due to secondary injuries in addition to the factors previously mentioned. As well as protecting the wound from friction, it is essential to prevent damage to the freshly stretched epidermis and granulation tissue during wound cleaning and dressing changes.

There is a need for materials that, once in place, do not require replacement and that are highly inductive in the tissue regeneration process. The materials are applied to fill the space in the skin defect after debridement, and then they are gradually degraded and absorbed by migrating cells from the surrounding tissue and replaced by autologous tissue. Early attempts have been made to develop versatile wound-dressing materials adapted to various wound types, from surgical injuries to chronic inflammation. Many approaches have been used in clinical practice, from traditional gauze and urethane foam as wound protection materials to chitosan- and collagen-based fiber sheets and autologous cell sheets made from patient cells [3]. Compared to skin grafting, which was previously the primary surgical treatment method, the problems of limited donor availability and low skin-grafting rates have been avoided; however, the complete regeneration of skin defects has not yet been achieved. Particular challenges are the low ability to induce tissue regeneration in chronic wounds and the difficulty of using the product on infected wounds. Although achieving the formation of new tissue that spatially compensates for defects, it was not possible to simultaneously achieve aesthetics, function, and the recovery of body hair, as in the case of the original living tissue. 

Therefore, in recent years, attention has been focused on tissue engineering approaches that induce skin regeneration by utilizing the tissue regenerative capabilities of the living body [4,5,6,7]. In the field of tissue engineering, the approach mainly focuses on the following three elements: cells that actively create and remodel neotissue; soluble factors (e.g., growth factors (GFs)) that regulate cell function; and scaffold materials that provide a mechanobiological tissue repair environment [3]. In the development of scaffold materials, the focus has been on maximizing cellular functions related to tissue regeneration rather than viewing scaffolds as mere substrates to which cells are attached and cultured. It attempts to improve cell behavior through cell–cell or cell–extracellular matrix (ECM) interactions by providing a skin-like healing environment for the cells involved in skin regeneration. Extensive research has been conducted to mimic the ECM of skin tissue in terms of composition and morphology. The perception of the environment by cells ranges from the physical and chemical properties of the substrate material to which they adhere to the mechanical properties. As summarized in these references, the roughness, charge, and hydrophilicity of the substrate surface affect the distribution of blood proteins adsorbed on the material surface and, together with the presence or absence of adhesive domains, have a strong influence on cell adhesion. In addition, as discussed in later sections, cells perceive the stiffness and spatial size of the adherent substrate as a mechanical stimulus, so it is also necessary to set the size and stiffness similar to those of the ECM of the tissue from which it is derived. These strategies can be broadly categorized into the use of ECM components (natural polymers such as collagen and elastin) for material selection, scaffold shapes of porous forms (fiber sheets, hydrogels, and sponges) that mimic the three-dimensional network structure of the ECM, and multilayered forms, such as the epidermis and dermis layers [2,8,9,10,11]. Moreover, the introduction of cells, GFs, and drugs selected with a focus on each stage of the wound-healing process, combined with these perspectives, makes it possible to increase the rate of neotissue formation [12]. Many dressings already in use clinically are in the form of fibers or sponges. Many investigations on the fabrication conditions for the fiber diameter and pore size in networks have been reported [13,14]. This has resulted in a more microscopic approach to material design in terms of cells’ perception of the surrounding environment.

Thus, the three-dimensional structure of the ECM, the environment surrounding the cell, and the use of bioactive substances such as GFs to mimic biological functions have made great strides. As discussed in later sections, many results have been reported in chronic wound models, such as diabetes and burns [15]. Therefore, the next challenge in cutaneous wound healing is to restore the hierarchy found in native skin with its original function. This review summarizes the wound-healing mechanisms in skin tissue and the research trends in 3D porous materials, such as fiber sheets and sponges, which are increasingly used in clinical applications. Research advances in developing materials as microenvironments on a microscopic scale are also discussed.

## 2. Skin Regeneration

### 2.1. Skin Structure

The skin is the largest organ and consists of three layers: the epidermis, dermis, and subcutaneous tissue (Figure 1) [16,17]. It is the outermost layer of the body and acts as a protective barrier against physical (e.g., external forces, dryness, ultraviolet rays, and temperature changes), chemical, and pathological insults (e.g., infection). It is constantly renewed from the inside to the outside, and the replacement of cells assists the immune system. The skin is equipped with several appendages; sweat glands play an important role in regulating body temperature, and hair, together with many nerves, serves as a tactile organ.

The epidermis is a stratified squamous epithelium derived from the ectoderm [18,19]. The epidermis is divided into five layers, which, from the bottom, are the basal layer, spinous layer, granular layer, light layer, and stratum corneum, with melanocytes mainly in the basal layer. Keratinocytes that divide and proliferate in the basal layer migrate to the upper layers during differentiation, resulting in keratinization, which involves denucleation and shedding from the surface of the stratum corneum. This skin turnover and the strong intercellular junctions between keratinocytes provide a barrier that prevents the entry of foreign substances and the loss of water from the body. In addition, the epidermal basement membrane, mainly composed of type IV collagen and laminin produced by the basement layer cells, not only tightly binds to the epidermis and dermis but also regulates the transport of substances across this boundary.

The dermis and hypodermis are connective tissues of mesenchymal origin. The dermis is a thick, dense connective tissue with dense collagen fibers, whereas the hypodermis is a sparse connective tissue containing large numbers of fat cells. The reticular layer, which constitutes the main part of the dermis, is composed of rough collagen fibers, whereas elastic fibers form a delicate meshwork on the surface of the dermis, leading to the basement membrane. The majority of cells in the dermis are fibroblasts; however, migrating cells, such as macrophages and lymphocytes, are also present. In the hypodermis, collagen and elastic fibers are oriented in various directions: horizontally in areas of the skin with a large range of motion and vertically in areas of the skin that are less mobile, such as the head and palms of the hands. On the surface of the hypodermis, a network of thin arteries extends into the subcutaneous tissue and dermis, and thinner arterial branches reach the surface of the dermis. Veins are present in a network in the upper dermis and superficial layers of the hypodermis and are involved in the regulation of body temperature.

### 2.2. Wound-Healing Processes

The definition of a wound is the disruption of the anatomical structure and function [20]. Acute wounds often result from burns, trauma, and surgery. Under normal circumstances, the skin has an excellent healing ability to regenerate tissue and restore function, with acute wounds generally healing between 8 to 12 weeks [21,22]. However, in cases of extensive and deep wounds that extend into the dermis layer and chronic inflammation, such as bedsores, burns, and diabetic ulcers, the healing process is often delayed because the ability to regenerate tissue is impaired [12].

Wound healing is a complex, multifactorial process that requires the intricate coordination of many types of cells and a permissive microenvironment (Figure 2) [2,23]. Acute skin wounds heal in four stages: hemostasis, inflammation, proliferation, and maturation [24,25]. After injury, platelets immediately appear in the blood aggregate and form a blood clot. Following hemostasis, the inflammatory phase occurs. Inflammatory cells, such as leukocytes, macrophages, and platelets, accumulate along with blood in the wound [26]. Inflammatory cells phagocytose bacteria from the wound surface, and macrophages secrete proteolytic enzymes that dissolve foreign substances on the wound surface to form a clean surface. The type, quantity, growth rate, and activity of cells that collect in a wound are all regulated by GFs. Therefore, an imbalance in GFs is one of the factors that perpetuate the inflammatory phase, and, along with wound infection, hypoxia, and impaired circulation, interferes with the progression of the normal healing process. The proliferative phase begins at the end of the inflammatory phase and progresses through the first month after injury. Fibroblasts and endothelial cells that migrate to the wound following the release of GFs during the inflammatory phase divide and form granulation tissue. Granulation tissue has abundant blood flow and active cell proliferation and is resistant to infection, protects the wound surface, and fills in tissue defects. It is composed of fibers, such as collagen, produced by fibroblasts, and neocapillaries produced by endothelial cells, which are used as scaffolds for the migration and division of epidermal cells to form a normal dermis layer around the wound from the wound margin to the center of the wound surface [27]. The final stage is the maturation phase, which may take several years to complete. The granulation tissue formed during the proliferative phase is remodeled and capillaries regress, forming dense scar tissue. This includes the maturation of collagen fibers. The scar tissue, which was initially more prominent than the surrounding normal skin tissue, approaches the strength of normal skin and becomes less prominent by the end of a 1-year period.

If this series of wound-healing processes does not progress, the wound becomes chronic and does not reach the final stage of healing. In many cases, the inflammatory phase is prolonged by an imbalance of inflammatory and anti-inflammatory cytokines, which impairs the subsequent transition to the proliferative phase. Cellular function is reduced in chronic wounds, with decreased collagen production in fibroblasts and the hyperproliferation and decreased migration of keratinocytes being reported [28,29]. The removal of these barriers to healing and establishing a pro-healing environment is necessary to promote wound healing in chronic skin injuries. The control of infection is often important, and in the case of pressure ulcers, attention must also be paid to tissue damage and the exacerbation of inflammation due to free radical generation from ischemia–reperfusion. These processes, in the case of chronic wounds, can be influenced by local or systemic factors: infection of the wound, hypoxia, impaired circulation, repeated wound damage, nutritional deficiencies (protein, vitamin, and mineral deficiencies), and aging, which reduces the healing capacity.

**Figure 2 biomimetics-08-00130-f002:**
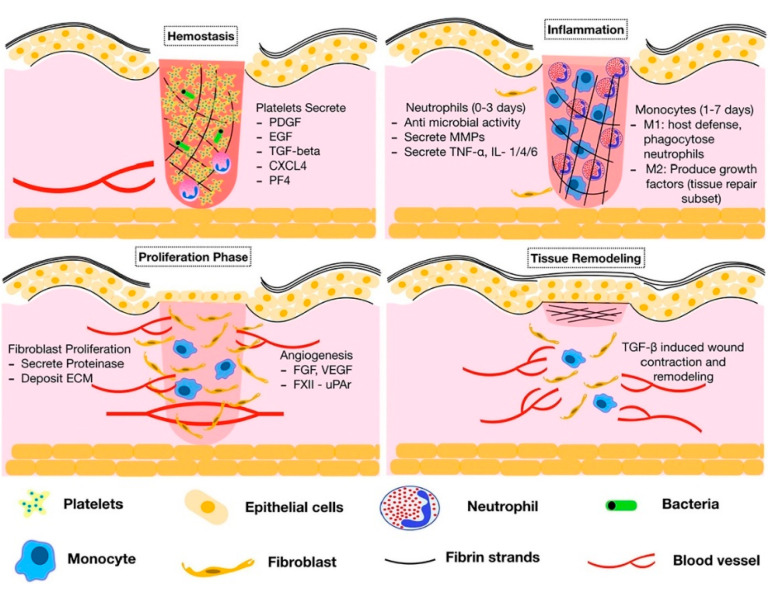
Wound-healing stages (hemostasis, inflammation, proliferation, and remodeling) and their major processes and components. Reprinted from [30] with permission from Elsevier.

## 3. Mimicking ECM Structure and Multilayer Nature of Skin

As already seen in commercially available dressings, the attention to 3D porous materials remains high. In addition to physical protection through the spatial complementation of skin defects, the continuous pores of the scaffolds are effective for the infiltration of cells and microvessels from the surrounding tissue, ECM deposition, and the exchange of oxygen and nutrients [14]. In this section, we will summarize the characteristics of material design and material selection for each of the nanofiber, sponge, and hydrogel morphologies. The porosity, pore size, and degradation period can be controlled for all of these materials, and multilayered materials with composites of materials with different properties are likely to increase in the future. Increased material selectivity according to the wound type is important not only because of its wound-healing effect but also because it allows the use of materials that are appropriate for the geometry of the wound. However, if one considers that the skin structure is divided into epidermal and dermal layers and, in particular, the importance of the barrier function of the epidermal layer against infection and desiccation, the usefulness of material design strategies specific to the epidermal layer becomes apparent. As manufacturing methods evolve, there are increasing attempts to combine nanofibers and meshes into sponges and 3D-printed structures. However, multilayered materials with widely varying mechanical properties, density, and connectivity can lead to delamination between structures. The evaluation of tissue infiltration and mechanical property changes associated with material degradation, both in vitro and in vivo, will help optimize composite design requirements.

### 3.1. Nanofibers

The electrospinning (ESP) method can form sheets of tiny fibers stacked in the range of a few nanometers to micrometers [31,32]. The sample solution in the spinning nozzle is affected by charge repulsion when a high voltage is applied, and this is collected on a collector. By using a rotating roller as the collector, it is possible to fabricate not only flat surfaces but also fiber-oriented sheets [32,33,34]. ESP may produce smaller fiber diameters than knitted fiber mesh, and non-woven ESP sheets (also called fabrics of felts) can be produced at a thickness of several hundred micrometers. The high surface-area-to-volume ratio, the strength of the fiber architecture, and the pore size mimic a biological ECM suitable for cell adhesion and proliferation [32,35,36]. The fiber diameter, fiber-to-fiber voids (pore size), and sheet thickness can be adjusted by changing the dispensing conditions: viscosity and conductivity of the sample solution, dispensing speed, dispensing distance, and electric field strength [37]. Randomly oriented nanofibers and the pores present between the fibers act as a physical barrier against infection [38]. On the other hand, intricate pores reduce cell infiltration compared to sponges and other 3D porous materials. In response, increased fiber diameters and the use of porogens have been used to further improve the pore size [37,39,40,41].

As base materials, biogenic polymers such as collagen, elastin, elastin-like peptides, silk fibroin (SF), and chitosan are often used due to their cell affinity and hydrophilicity, while synthetic polymers such as polyethylene glycol (PEG), poly (ε-caprolactone) (PCL), and polyvinyl alcohol (PVA) polymers are also used [38,42,43]. Blends of several polymers are often used to allow compatibility at the micro level, facilitate fiber formation, and control degradation behavior and mechanical properties. Using the coaxial nanofiber method, Lan et al. fabricated sheets composed of coaxial bilayer nanofibers using PCL as the shell layer and PVA as the core [44]. By adding antibacterial ε-poly (L-lysine) (ε-PL) to the shell layer and antioxidant tea polyphenols (TPs) to the core, they aimed to obtain bacterial inhibition in the early stages of healing and longer-term antioxidant activity. Ju et al. focused on the non-inflammatory properties of SF and its ability to promote cell adhesion and proliferation, including keratinocytes, and prepared ESP sheets with expanded bulk volume and pore size by employing natrium chloride crystals as a porogen (Figure 3). The SF/PEO solution mixed with poly(ethylene oxide) (PEO) to improve spinnability was dispensed onto a collector carrying sodium chloride crystals to obtain an SF nanomatrix by the recrystallization process of SF and the leaching of sodium chloride particles and PEO. The speed of wound closure and the formation of new tissue were evaluated in a deep second-degree burn model in rats, using Medifoam^®^ (polyurethane hydrocellular dressing foam) as a control. The SF nanomatrix group showed faster wound closure than the Medifoam^®^ group and regenerated to a morphology comparable to normal skin on day 14. In addition, uniaxially oriented fibers can orient the arrangement of cells cultured on the sheet and promote migration [45,46].

Despite this similarity to ECM-like structures, the use of ESP sheets in cutaneous wounds has been challenging in terms of macroscopic handling. The wound shape and depth vary from patient to patient, making it difficult to fit them into the wound. The recent advent of portable ESP devices has made in situ fabrication possible, and Yue et al. have developed a lightweight device capable of using high voltages of up to 11 kV at perfusion rates ranging from 0.05 mL/h to 10 mL/h [47]. Xu et al. mixed a PVA solution with bone-marrow-derived stem cells (BMSCs) and spun them directly into rat full-thickness skin wounds. BMSCs have been reported to promote re-epithelialization and dermal neogenesis in chronic wounds; PVA/BMSC sheets maintained >90% cell viability and showed accelerated wound closure while acquiring a thick epithelial layer after 7 days. Further mechanical improvements will promote the application of integrated biological DDS systems, including cells and GFs [45,48].

### 3.2. Sponges

Sponges are often selected among porous materials because of their relatively large pore size, high porosity, and high water content capacity. Sponges may have varying pore sizes depending on the fabrication method. Relatively small pore sizes (20–160 μm) can be formed for spherical particles, whereas other methods, such as using porogens or gas foaming, can obtain a wide range of pore sizes of hundred micrometers [14]. Their pores are often discontinuous or have bridged ends, and their shape, density, uniformity, and permeability affect mechanical properties, water-containing capacity, and cell infiltration properties. A frequently used method of sponge preparation in the field of tissue engineering is the lyophilization of aqueous solutions. For sponges with microscale pores, the removal of such sacrificial materials, called porogens, is chosen for its simplicity. Combinations of porogens and their removal methods include ice and lyophilization and removal by dissolving salts or spherical particles in a solvent [14,49,50,51]. There is also the gas foaming method, which avoids the use of organic solvents. Supercritical carbon dioxide is used to form pores as the polymer solution expands back to its gaseous state. Due to limitations imposed by the solubility of carbon dioxide, surfactants and water emulsions are used, which affect pore size.

Collagen in particular is often chosen as a base material, and antibacterial chitosan and SF, which have been shown to promote healing, have also been used [7,52,53]. Another attractive feature is its high water content and the ease of adding a DDS, such as adding it to a solution or impregnating it after sponge formation [54]. Sponges obtained by the lyophilization of a collagen/chitosan solution were immersed in heparin solution with the crosslinker 1-ethyl-(dimethylaminopropyl) carbodiimide/N-hydroxy sulfosuccinimide and then underwent secondary lyophilization. The sponge showed sustained heparin release in vitro [53,55]. In a guinea pig second-degree burn model, it was suggested that in addition to the wound-healing-promoting effects of collagen and chitosan, heparin formed a complex with GF, leading to accelerated wound closure. Moreover, a sponge crosslinked with keratin and alginate induced wound closure by re-epithelialization more rapidly than a standard collagen sponge in a porcine burn model [56]. More notably, tissue staining confirmed the expression of type IV collagen in the papillary layer, which achieved basement membrane regeneration. This latter design is attractive because it showed healing effects on chronic wounds without the use of exogenous GFs.

### 3.3. Hydrogels

A hydrogel is a three-dimensional network structure composed primarily of macromolecules in a water solvent [8]. This network is formed through covalent or non-covalent bonds, which vary depending on the presence or absence of non-covalent interaction points or chemical crosslinking points based on the structures of the biopolymers and synthetic polymers used. Hydrogels are relatively easy to use in situ compared to other morphological processing methods. In particular, biopolymers such as collagen and elastin mimic the spatial and biological environment of the skin ECM around the cells [8]. The mechanical properties, degradation period, and DDS behavior can be controlled not only by the choice of material but also by the choice of the physicochemical crosslinking method and its crosslinking density. In addition to collagen, elastin, and SF, which are used as well as other forms of biomacromolecules, other natural polymers are often selected, such as hyaluronic acid and other proteoglycans, which focused on skin ECM components, alginic acid, which has a long track record of use, and chitosan, which is used to obtain antibacterial properties.

To induce dermal regeneration and angiogenesis with high efficiency, a composite hydrogel incorporating bFGF into carboxymethyl chitosan (CMCS)-PVA was prepared. It showed the sustained release of bFGF for 2 weeks in vitro and achieved improved dermal neogenesis and re-epithelialization rates as well as neovascular density in full-thickness burn wounds in rats [57]. Furthermore, a chitosan veil hydrogel was developed for application to chronic wounds such as ulcers and bedsores, which have been slow to adapt to other wound types [58]. Reversible intramolecular physical interactions through dynamic hydrogen bonding with the introduction of PEG can impart the self-healing ability of hydrogels. The repair of hydrogels from breakdown caused by the movement of the human body opens the possibility of their application to chronic wounds with a long healing period. Evaluation in an ex vivo human ulceration model has shown accelerated re-epithelialization and dermal tissue remodeling, and the addition of antibiotics and non-steroidal anti-inflammatory drugs (NSAIDs) may strongly induce chronic wound healing.

In addition, the use of photopolymerization is increasing, as the skin is the body’s outermost layer [59,60]. The combination of pectin methacrylate (PECMA) and gelatin methacryloyl (GelMA) has led to the development of injectable hemostatic hydrogels [61]. Evaluation in an in vitro porcine skin bleeding model showed that direct injection into the wound and rapid photocrosslinking reduced the coagulation time by approximately 40% and then allowed for easy removal after use. Photocrosslinked GelMA hydrogels with red jujube powder, which has antioxidant activity, were used in a rat skin model of total skin defects, and a marked enhancement of wound closure was observed [62].

Recently, synthetic hydrogel matrices have begun to mimic basement membranes [19,63]. Of the hierarchical structures of the skin, the basement membrane is one whose importance has been increasingly recognized in recent years. Both cell–cell and cell–ECM interactions are essential for the proper expression of cellular functions. The materials of choice include the natural basement membrane components type IV collagen and laminin, as well as purified type I collagen, fibrin, alginate, and hyaluronic acid. In addition to mimicking the topography, as described in later sections, the focus is on the uptake of the biochemical properties of the basement membrane. Integrin-mediated cell–ECM interactions, microenvironment degradation and reconstruction by cells, and GF-mediated processes of epithelial or endothelial cell adhesion, proliferation, differentiation, and migration proceed [63]. These will receive more attention in the future as the mechanisms of epithelial morphogenesis and the role of the microenvironment in reprogramming using iPSCs are better understood.

### 3.4. Composite Materials

As seen in commercial products, composites of materials with different morphologies are often used to achieve both the physical barrier function against infection and foreign invasion that the epidermal layer of the skin provides and the progressive neoformation and reconstruction of the dermal layer [64,65]. In the epidermal layer, meshes, ESP sheets, and films tend to be the materials of choice [66,67]. Miguel et al. used an ESP sheet composed of SF/PCL as the top layer to mimic the dense architecture and water resistance of the epidermal layer, spinning it to a thickness of 0.05 to 1.5 mm, comparable to that of the human epidermal layer. An SF/hyaluronic acid blend solution corresponding to the dermal layer was added to thymol, which has antimicrobial activity, and ESP was spun on the epidermis-layer sheet to a thickness of 1.5 to 4 mm of the human dermis layer and then created a bilayer composite material (Figure 4a,b) [38]. Although SEM observations showed morphological differences between the two layers, they did not detach during the in vitro degradation period. In addition, when evaluated for antibacterial activity against Gram-positive and Gram-negative bacteria, bacteria adhering to the upper-layer surface of the sheet were unable to infiltrate its underside, indicating the usefulness of the epidermis-layer sheet as a physical barrier against infection. Furthermore, the presence of thymol in the dermal-layer sheet inhibited bacterial growth. Thus, by mimicking physical and biological skin functions, it is expected that stronger and more valuable materials will be developed for the prevention of wound infection. Another strength of ESP nanofibers is that they can be dispensed onto the surface of a sponge or 3D-printed structure, which is the dermal-layer portion of the previously formed structure, to increase the adhesion of the boundary surfaces. Nanofibers can provide interconnectivity between microscale pores while maintaining high porosity and surface-to-volume ratios in mechanically stable structures. Electrospinning was performed on a mesh structure consisting of 3D-printed microfibers to orient the nanofibers in the gaps between the meshes (Figure 4c,d) [68]. Due to the thickness and conductivity of the 3D printing material, the bead structure may be more likely to form than when using only a collector as usual. However, it was possible to form a uniform dispersion of uniaxially oriented nanofibers between and within the micro-sized pores. The cultivation of fibroblasts on the material showed that the distribution of the cells spread over time, and after 14 days, the cells were elongated and aligned along the aligned nanofibers. In the latest report, methacrylated gelatin (MeGel) hydrogel containing the Salvia miltiorrhiza Bunge-Radix Puerariae herbal compound (SRHC) and radially oriented MeGel/poly(L-lactic acid) (PLLA) nanofiber sheets with radially oriented MeGel/poly(L-lactic acid) were prepared [69]. They were prepared by placing ESP nanofiber sheets in a mold and adding the MeGel hydrogel precursor solution before photocrosslinking. Fibroblasts added to the hydrogel were shown to migrate into the ESP nanofibers and align along the radial fiber orientation. In addition, wound closure and hair follicle regeneration were significantly increased by the addition of SRHC in a full-thickness skin defect model of type 1 diabetic mice. Other simple methods of coating the surface of a collagen sponge with a chitosan solution have been reported [70]. Thus, the variety of complexation methods is increasing.

## 4. Incorporation of Drug Delivery System (DDS) Functionality

A type of wound dressing that has been studied extensively in recent years is based on tissue engineering concepts, some of which are already in practical use. In recent years, as described in previous sections, the introduction of cells, growth factors, and other bioactive substances have been reported to improve healing ability by promoting epithelialization and tissue regeneration, as well as the usefulness of highly functional materials that have acquired antimicrobial properties. For more practical function, DDS properties should be considered for the sustained release of these substances over time or with material degradation.

### 4.1. GFs

Combining GFs with biocompatible materials was implemented earlier due to their potent effects, receptor-mediated promotion of cell proliferation, and differentiation for tissue regeneration [71]. The stability of GFs in vivo is a challenge due to their short half-lives, as basic fibroblast growth factor (bFGF) loses up to approximately 10% of its activity within 24 h of administration. Another limitation is the low cost-effectiveness due to diffusion and leakage from the wound. Therefore, there is a growing interest in designing controlled release rather than applying or spraying GFs directly to the wound. Pelnac Gplus^®^ produced by Gunze contains alkaline-treated gelatin in its sponge layer, which traps bFGF through electrostatic interactions. Furthermore, the sponge is degraded by collagenase produced by migrating fibroblasts, releasing bFGF slowly for more than 10 days [72]. The use of electrostatic interactions involves chemical modification and the incorporation of negatively charged glycosaminoglycans such as heparin [73,74,75]. Additionally, vascular endothelial growth factor (VEGF), platelet-derived growth factor (PDGF), and epidermal growth factor (EGF) are used specifically in wound healing to restore damaged skin [71,76,77].

### 4.2. Cells

Combination approaches using scaffolds and human cells (i.e., autologous cells, allogeneic cells, and induced pluripotent stem cells (iPSCs)) have increased, particularly in the last decade. Cells can be seeded directly into materials, encapsulated and encapsulated, or decellularized after culturing on materials to obtain ECM components, among other approaches. Autologous cultured cell sheets have been used to treat severe burns and congenital vitiligo by culturing cells in sheet form and transplanting them. However, the limited quantity and quality of cells that can be harvested from donors, the limited culture time for emergency use, the high cost of production, and challenges with post-transplantation engraftment rates have limited the number of patients for whom this technology can be used. Therefore, research is expanding toward using human-derived cells, for example, not only to create cell-only structures but also to use materials to provide structural support and a healing environment.

Endogenous mesenchymal stem cells (MSCs) are self-renewing pluripotent stem cells that can differentiate into a variety of tissues of mesenchymal origin, including bone and fat [78]. MSCs act in repair by modulating immune responses and inflammation, including the mobilization of other host cells and the secretion of GF and ECM proteins; MSCs secrete cytokines, chemokines, GFs (especially VEGF, PDGF, bFGF, EGF, keratinocyte growth factor, and transforming growth factor beta), exosomes, and other tissue repair mediators [79]. Cells involved in regeneration (e.g., keratinocytes, fibroblasts, and endothelial cells) respond to paracrine signaling from MSCs by altering gene expression related to proliferation and migration. In addition, they induce the suppression of proinflammatory cytokines and increase anti-inflammatory cytokines, and thus, the use of exogenous MSCs has shown a healing effect even in chronic wounds [80,81]. When adipose-derived MSCs (when exosomes are isolated from culture supernatants of adipose-derived stem cells) were loaded on alginate hydrogels, wound closure, collagen production, and angiogenesis were significantly improved in a rat skin model of all-layer defects [82].

## 5. Mechanobiological Approaches

In recent years, there has been renewed interest in mechanobiological research to elucidate the processes by which cells recognize their surrounding environments, such as other cells and base materials, and regulate cellular functions [83]. In conventional materials research, structural mimicry and the control of physicochemical properties on a macroscopic scale larger than that of cells, as described above, have mainly been investigated. If the cellular function can be modulated through the design of a finer-scale environment, the recognition of the chemomechanical scaffold environment by the cell has the potential to replace the use of multiple bioactive substances. The processes by which cells receive mechanical stimuli are now being elucidated, and mechano-dependent phenotypes include cell migration, proliferation, secretion, and differentiation. It has been reported that the formation of focal adhesion, a complex involved in integrin-mediated cell adhesion and migration, is regulated by both the ECM ligand distribution and substrate rigidity [84,85]. However, the actual biological environment is complex and involves many factors, such as interactions between multiple cell types and material properties that change over time, and there are many unresolved aspects. Materials and models with structural properties similar to those of living tissues have been used to elucidate cell-specific functional regulatory processes. In a recent study, treatment of the mechanotransduction signaling pathway with a small-molecule focal adhesion kinase inhibitor in a porcine model of autologous segmental skin grafts resulted in a shift toward anti-inflammation and induced fibroblasts to a differentiated fate, which led to decreased contractures, alleviated scar formation, and reconstructed collagen structure. This ultimately resulted in the transplanted skin acquiring biomechanical properties similar to those of the normal tissue [86].

In the analysis of cell behavior and gene expression changes induced by materials, patterned scaffolds made of synthetic polymers are often used because of their ability to form dense and complex phase structures. Sima et al. fabricated grid-patterned materials with different line widths (30 to 100 μm) by photopolymerization via two-photon polymerization and evaluated cell behavior [87]. Human dermal fibroblasts (HDFs) and human epidermal melanocytes showed adhesion and proliferation oriented to the lines of the scaffold. HDFs with larger cell sizes were only oriented toward thicker line widths. This orientation to the substrate architecture has also been observed in other cell types, such as mesenchymal stem cells and neuronal cells [88,89]. In particular, fibroblast alignment helps to suppress scar tissue and obtain highly anisotropic collagen deposition and ECM [90]. Recently, changes in cellular function in macroscopic patterning materials that mimic the characteristic undulating structure of biological skin tissue have also been studied. Mobasseri et al. cultured keratinocytes on a material that mimicked the three-dimensional undulating structure of the basement membrane, which separated the epidermal and dermal layers of human skin tissue, and evaluated the effect of the undulating location of the stem cells on their differentiation characteristics (Figure 5) [91,92]. After seeding cells on the swell pattern, differentiated cells were localized at the foot of the swell. β1-Integrin-positive and flattened shapes were observed at the tip of the swell. The process by which cells perceive the phasing of a material as a mechanical stimulus was suggested and initiated by attractive forces generated by intercellular adhesion. Further elucidation of the effects of the micro-surface topography of the material on cellular function and its application to material design may lead to the restoration of a multilayered structure of the skin that more closely resembles that of normal tissue.

## 6. Dressings in Clinical Usage

The understanding of clinical healing has led to the development of a variety of wound dressings tailored to characteristics such as wound location, size, depth, and the presence of infection. Ghomi et al. summarized the process of selecting a dressing based on the wound depth, exudate volume, and wound characteristics (Figure 6) [4].

Products that have already reached clinical use allow for the induction of tissue regeneration in addition to the traditional protection against dryness and secondary physical damage. In burns and relatively small acute traumatic wounds, there are already effective artificial dermis, skin substitutes, and other dressings, as shown in Table 1 [5,93]. However, it is still often difficult to restore the multilayered structure and aesthetics of the skin, including the appendages. Furthermore, long-term care for extensive burns, infected wounds, and chronic wounds remains challenging [2].

One of the characteristics of many of the products in practical use is that they are made of materials that have a proven track record in the medical field and/or have a porous form. These materials include natural biopolymers (i.e., collagen, gelatin, elastin, and glycosaminoglycans) composing the ECM of the skin, as well as synthetic polymers (i.e., silicone). Like other materials for blood vessel and bone regeneration, scaffolds need to be bioresorbable and allow for cell infiltration and ECM production. In addition, since biomaterials are used in contact with body fluids at the boundary with the living body, biocompatibility is a requirement so as not to cause excessive inflammatory reactions that might hinder the wound-healing process. Collagen is often selected because it can be obtained relatively inexpensively from pigs and other sources, and its integrin-mediated cell adhesion supports cell migration to and proliferation in the wound. In terms of form, films, foams, and gels are frequently selected for their water retention properties and conformity to the wound shape. In addition, there are two-layer materials such as a silicone layer/collagen sponge that focus on the skin’s large layered structure of the epidermis and dermis. Such multilayered materials are useful for both the biological barrier function, which is primarily performed by the epidermal layer of the skin, and the formation of broad and thick neoplastic tissue. The breathability and moderate moisture content of the material keep the wound moist but prevent the maceration of the surrounding normal tissue due to excess moisture. This helps in autolytic debridement during the healing process by inflammatory cells [2]. Several tissue engineering materials are already being used for skin regeneration and are being explored for further application with the addition of GFs and cells [72,106]. Similarly, nanofibrous sheets and sponges based on bio-derived polymers such as collagen, chitosan, and silk are increasingly being investigated for use in chronic wounds with the addition of antimicrobial agents [107,108,109,110].

## 7. Conclusions

Currently, available dressing approaches for skin wound healing are becoming increasingly diverse. However, materials with enhanced functionality are needed to address delays in infection and tissue regeneration, especially in chronic wounds. The fabrication methods of 3D structures and the selection and combination of GFs and drugs to mimic the structure and biochemical function of the ECM are being investigated. However, there is room for improvement in the combination of fabrication techniques of materials as physical barriers against infection and the selection of antimicrobial drugs and/or components. The inhibition of bacterial invasion and clearance must be modulated so that it does not interfere with healing to be effective in chronic ulcers, where the body’s immune mechanisms are compromised. The use of exogenous cells and GFs has a strong effect on tissue regeneration but can also be a barrier to the development of more versatile materials. Recognition of the surrounding microenvironment by cells has received renewed attention in recent years. Recognition of the structural and mechanical properties of materials is important for controlling long-term cell behavior because these properties alter cellular function. Cells have been shown to recognize a variety of material morphologies and structures, from the nanoscale to the macroscale, which are smaller than the cell size. In the future, the elucidation of the recognition mechanism of mechanical stimuli based on factors such as cell–cell interactions in material structures that mimic the environment of living organisms will help define requirements such as the scale of structures to focus on in material design. In the future, it is hoped that a combination of these strategies will be used and the contribution of each will be reevaluated as the healing period progresses, moving the stage toward the development of a comprehensive and balanced dressing.

## Figures and Tables

**Figure 1 biomimetics-08-00130-f001:**
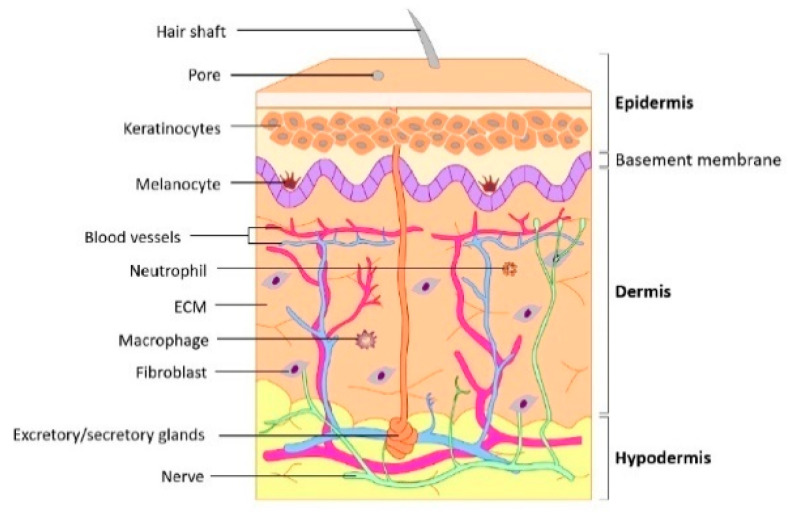
Schematic representation of human skin and underlying structures. Reprinted from [17] with permission from Elsevier.

**Figure 3 biomimetics-08-00130-f003:**
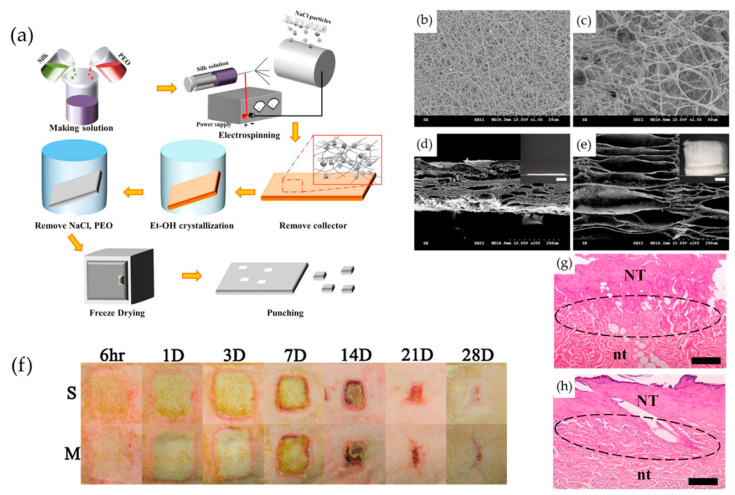
(**a**) A schematic depiction of the fabrication of the electrospun SF nanomatrix. SEM images of the (**b**,**c**) surface and (**d**,**e**) cross-section view of (**b**,**d**) the SF nanosheet and (**c**,**e**) SF nanomatrix with salt particles poured during electrospinning. (**f**) Closure of wound treated with S (SF nanomatrix) and M (Medifoam^®^). H&E staining images of burn wound tissues at 7 days after treatment with (**g**) SF nanomatrix and (**h**) Medifoam^®^. (NT: necrotic tissue; nt: normal tissue; circle: keratinocytes; scale: 50 μm.) Reprinted from [40] with permission from Elsevier.

**Figure 4 biomimetics-08-00130-f004:**
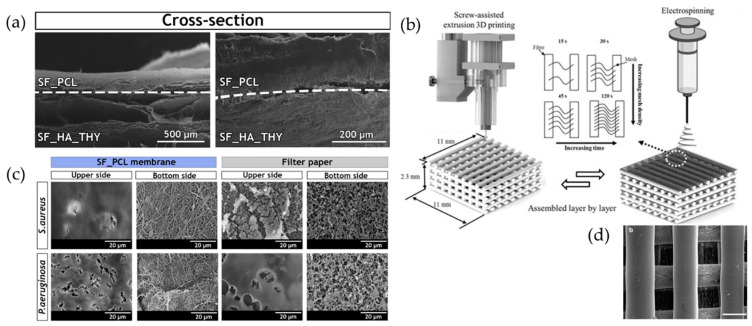
SEM images of (**a**) EAM cross-section (top layer: SF_PCL; bottom layer: SF_HA) and (**b**) microorganisms (*S. aureus* and *P. aeruginosa*) that adhered to the upper or lower side of the SF_PCL membrane and filter paper (control group). Reprinted from [38] with permission from Elsevier. (**c**) Schematic process of a dual-scale scaffold with 3D printing and electrospinning. (**d**) SEM image of the dual-scale scaffold with electrospun (45 s) nanofibers (scale bar = 300 μm).

**Figure 5 biomimetics-08-00130-f005:**
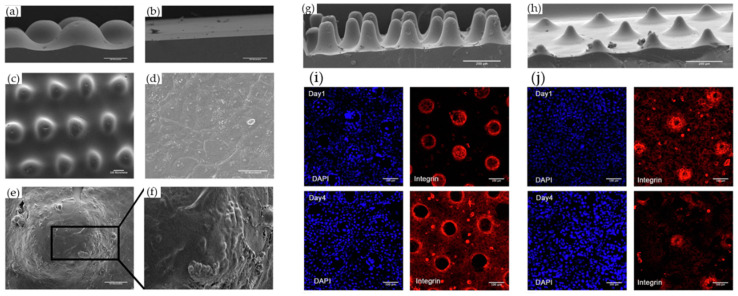
(**a**–**f**) SEM images of keratinocytes cultured on PDMS substrates before collagen coating: side views of (**a**) patterned and (**b**) flat, and (**c**) top view of patterned. (**d**–**f**) Keratinocytes on (**d**) flat and (**e**,**f**) undulating PDMS substrates. (**f**) A higher magnification view of the boxed region in (**e**). Scale bars: 100 μm (**a**–**c**), 50 μm (**d**,**e**). (**g**–**j**) Effect of culturing keratinocytes on topographies with different slopes. (**g**,**h**) SEM images of uncoated patterned PDMS with (**a**) steep and (**b**) shallower slopes. (**i**,**f**) Images of keratinocytes immunolabelled for β1 integrin (red) with DAPI counterstain (blue) with (**i**) steep and (**j**) shallower slopes. Scale bars: (**g**,**h**) 200 μm (**i**,**j**), 100 μm. Reprinted from [92] with permission from Elsevier.

**Figure 6 biomimetics-08-00130-f006:**
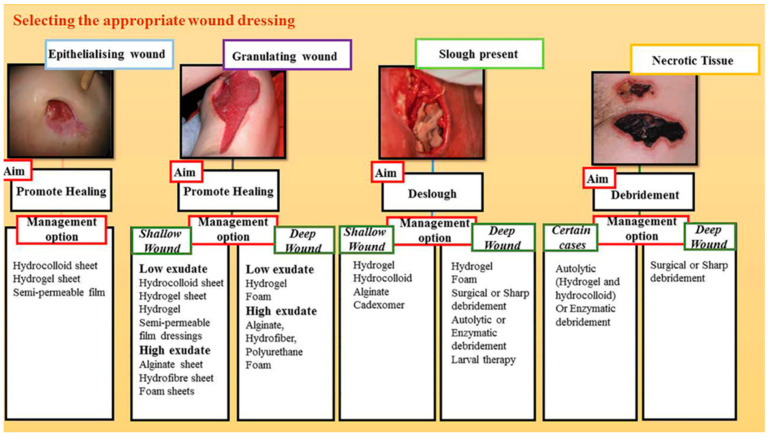
Effective parameters for selecting the appropriate wound dressing. Images reprinted from [4] with permission from ©WILEY-VCH Verlag GmbH & Co. KGaA, Weinheim, Germany.

**Table 1 biomimetics-08-00130-t001:** List of wound dressings in the world markets.

Dressing	Types	Form	Major Components	Ref.
Dermagraft^TM^	Dermal substitute	Hybrid	Polyglactin mesh, human dermal fibroblast	[94]
Apligraf^TM^	Epidermal and dermal skin substitutes	Bilayer	Cell	[95]
Integra^TM^	Artificial skin	Hybrid	Bovine tendon collagen /glycosaminoglycan, silicone layer	[96,97]
AlloDerm^®^	Acellular dermal matrix	Decellularized tissue	Human dermis	[98]
Stratice™	Acellular dermal matrix	Decellularized tissue	Porcine dermis	[99]
SurgiMend^®^	Acellular dermal matrix	Decellularized tissue	Bovine dermis	[99]
Transcyte^®^	Tissue-engineered skin substitute	Hybrid	Nylon mesh, silastic layer, human fibroblast	[98]
Genocel^®^	Tissue-engineered skin substitute	Nanofiber (Solution-blow)	Gelatin	[100]
Pelnac^®^	Artificial dermis	Hybrid	Silicone layer, collagen sponge	[101]
Beschitin-F^®^	Skin substitute	Hybrid	Chitin-coated gauze	[102]
Matriderm^®^	Dermal substitute	Three-dimensional matrix	Bovine type I collagen, elastin	[103,104,105]

## Data Availability

Not applicable.

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
