# Peer review of "Recent Tissue Engineering Approaches to Mimicking the Extracellular Matrix Structure for Skin Regeneration"

_biomimetics, 2023, doi:10.3390/biomimetics8010130_

Round 1

Reviewer 1 Report

This is an interesting paper and it should be published given the authors address the following issues:

1. A more detailed description of the materials used for the process of tissue regeneration.

2. The physical and chemical properties as well as the mechanical properties need to be discussed though the impact that they have. 

3. What is the major take home message? It seems to be rather general. The authors should strengthen the conclusions and the message of the paper.

Author Response

We appreciate your kind and highly constructive comments on our manuscript. Please see the attachment. 

Reviewer 2 Report

The review by Hama et al. gives an overview on the advances in material design strategies for skin tissue regeneration application. Some major revisions should be performed before publication.

1. The authors are recommended to give the number of papers and patents (a graph could be added) in this field in the introduction in order to justify a review in this topic. The keywords used to make this graph should be added in the figure legend.

2. Some recent reviews in the area of  material design strategies for wound healing application like 10.1007/s11706-021-0540-1, and 10.3390/nano12050784 are suggested to be mentioned in the introduction. A justification for an additional review in this field should be given.

3. A graphical abstract that can reflect the scope of the review well is recommended to be drawn and presented in this paper, which will be helpful for understanding of readers.

4. In the section 2.2, some more descriptions about the healing of chronic wounds like diabetic wounds and bedsores are suggested be added.

5. In the section 3, the merits and demerits of different strategies are suggested to be summarized.

6. A table which summarize the combined application of bioactive molecules and biomaterial scaffolds should be added in the section 4.

7. Some recent works based the advanced development of composite materials for skin tissue regeneration like 10.1016/j.apmt.2022.101542 and 10.1016/j.jtv.2021.09.003 are suggested to be discussed in Section 3.4.

8. Some statements feel they are lacking references. Admittedly, some of these statement might be considered well known facts, the concepts mentioned might have been referenced previously or will be in the future, but it might still be pertinent add references next to these statements for new readers to the field, skim readers or people who don't necessarily want to go looking for the relevant reference.

9. The authors should make a critical review instead of plain text flow. This means that a comparative discussion should take place assisted with categorization of the attributes of the different systems in each section.  

10. The current limitation and challenges and future prospect should be fully discussed in Section 7.

11. The paper contains some typo and graphical errors. Please read carefully and correct Them.

Author Response

(The authors gave the same response as above.)

Round 2

Reviewer 2 Report

The authors addressed most of the reviewer's comments. Please double check the place of Figure 5 and 6. And the language is suggested to be carefully polished.